# Efficient Small Object Detection You Only Look Once: A Small Object Detection Algorithm for Aerial Images

**DOI:** 10.3390/s24217067

**Published:** 2024-11-02

**Authors:** Jie Luo, Zhicheng Liu, Yibo Wang, Ao Tang, Huahong Zuo, Ping Han

**Affiliations:** 1School of Information Engineering, Wuhan University of Technology, Wuhan 430070, China; ajam525@whut.edu.cn (J.L.);; 2Wuhan Chuyan Information Technology Co., Ltd., Wuhan 430030, China; 312275@whut.edu.cn

**Keywords:** aerial images, small object detection, RepNIBMS module, WFPN module, tri-focal loss function

## Abstract

Aerial images have distinct characteristics, such as varying target scales, complex backgrounds, severe occlusion, small targets, and dense distribution. As a result, object detection in aerial images faces challenges like difficulty in extracting small target information and poor integration of spatial and semantic data. Moreover, existing object detection algorithms have a large number of parameters, posing a challenge for deployment on drones with limited hardware resources. We propose an efficient small-object YOLO detection model (ESOD-YOLO) based on YOLOv8n for Unmanned Aerial Vehicle (UAV) object detection. Firstly, we propose that the Reparameterized Multi-scale Inverted Blocks (RepNIBMS) module is implemented to replace the C2f module of the Yolov8n backbone extraction network to enhance the information extraction capability of small objects. Secondly, a cross-level multi-scale feature fusion structure, wave feature pyramid network (WFPN), is designed to enhance the model’s capacity to integrate spatial and semantic information. Meanwhile, a small-object detection head is incorporated to augment the model’s ability to identify small objects. Finally, a tri-focal loss function is proposed to address the issue of imbalanced samples in aerial images in a straightforward and effective manner. In the VisDrone2019 test set, when the input size is uniformly 640 × 640 pixels, the parameters of ESOD-YOLO are 4.46 M, and the average mean accuracy of detection reaches 29.3%, which is 3.6% higher than the baseline method YOLOv8n. Compared with other detection methods, it also achieves higher detection accuracy with lower parameters.

## 1. Introduction

In recent years, the development of drones has been rapid and significant. In the military field, drones have become a vital combat tool. Many countries are currently engaged in the active development of drone systems for a range of military purposes, including reconnaissance, target strikes, intelligence gathering, surveillance patrols, and other operations. In the civilian sector, the potential applications of drones extend to numerous fields, including transportation, agriculture, logistics, security, urban construction, disaster warning, etc. The majority of aerial images contain a considerable number of small objects. The Common Objects in Context (COCO) dataset [1] is a commonly utilized dataset for object detection, whose definition of object size is that objects with dimensions smaller than 32 × 32 pixels are classified as small objects, objects with dimensions between 32 × 32 pixels and 96 × 96 pixels are designated as medium objects, and objects with dimensions larger than 96 × 96 pixels are designated as large objects. As an illustration, the VisDrone dataset [2] exhibits a nearly two-thirds proportion of small objects, as demonstrated in Figure 1. Consequently, in the aforementioned fields, the rapid and accurate identification of small objects can markedly enhance operational efficiency. Hence, the algorithms for the detection of small objects in aerial images are of great significance.

Small object detection in aerial images encounters numerous challenges because of the uncertain operating environment and complex image acquisition of drones. Unconstrained flight altitude and angle result in small, densely distributed, and variably scaled objects, posing difficulties for identification and tracking. A complex and changing background with low contrast, along with elements like trees and buildings in natural environments, hampers detection. Moreover, lighting and weather fluctuations impact image quality. Strong or backlight leads to overexposure or detail loss, and rain blurs images. Consequently, algorithms must be capable of handling small, densely distributed objects.

In addition, the object detection algorithm for aerial images will eventually be deployed on drone terminal devices, which are predominantly constrained by their hardware and lack a robust graphics processing unit (GPU). Some existing complex deep learning models, although having a relatively high detection accuracy in theory, cannot run in real time on drones due to excessive parameters and a large number of calculations, which limits their value in practical drone applications. Consequently, the algorithm must consider its own complexity to ensure that it does not excessively occupy the limited available processing power. Furthermore, it must achieve an effective balance between accuracy and speed.

To address the challenges, we aim to enhance the detection performance of lightweight object detection algorithms. The You Only Look Once (YOLO) series of detection algorithms are popular real-time algorithms known for speed and efficiency. YOLO algorithms view object detection as a single regression problem, mapping original image pixels to bounding box coordinates and category probabilities. They have three main components: backbone (extracts feature information), neck (fuses the information), and head (predicts category and location). This design allows YOLO to operate at high speeds with good accuracy, suitable for real-time applications. YOLOv8n is a more streamlined object detection algorithm that meets the requirements of both accuracy and speed. The YOLOv8n model has a relatively small number of parameters (3 M) and floating-point operations (8.1 GFlops), which contributes to its overall compactness. The objective of this paper is to enhance the detection accuracy of YOLOv8n while preserving its lightweight attributes.

We present a novel, effective small object detection algorithm, ESOD-YOLO, for aerial images. The algorithm is specifically designed to achieve superior detection results on resource-constrained UAV platforms. In light of the shortcomings of YOLOv8n in the context of small object detection, which contains the inadequate extraction of features by the backbone, the insufficient fusion of features at different scales by the neck, and the poor detection of small objects by the detection head, we have enhanced the original model in the following key respects:The RepNIBMS module is proposed as a replacement for the C2f module of the YOLOv8n feature extraction network. This module is designed to enhance the model’s ability to process objects of varying scales while retaining as much useful information as possible. The internal RepConv structure is capable of optimizing the performance of the network during training, simultaneously reducing the model’s memory and running speed during inference;The WFPN cross-level multi-scale feature fusion module has been proposed and implemented as a solution to the issue of identifying and locating small objects, which are often challenging to discern due to their diminutive size and inconspicuous features. To this end, the neck component has been expanded to accommodate supplementary feature maps for subsequent small object detection;In the head section, a small object detection head is incorporated to identify minute objects within feature maps replete with rich semantic and spatial data;A straightforward and effective tri-focal loss function is put forth on the foundation of the initial boundary regression function, Complete Intersection over Union (CIOU). Focusing on different regression samples enhances the detection effect, and the issue of an imbalance between easy and difficult samples is effectively resolved. This approach ensures that the model allocates greater attention to difficult samples, thereby improving the regression effect of the model.

## 2. Related Work

### 2.1. Conventional Object Detection Algorithm

In the early stages of object detection, traditional methods were employed based on manually extracted features. Two examples are notable. One is the Viola–Jones detector [3], proposed by Viola P. and Jones M. Another is the HOG detector [4], proposed by Dalal et al. The traditional approach to object detection is associated with several drawbacks, including high computational complexity, slow processing speeds, the necessity for manual design of feature extraction methods, and poor generalization capabilities.

Following the introduction of AlexNet [5] in 2012, there was a shift in focus towards convolutional neural networks (CNNs).

The most prevalent categorization of CNN-based object detection methodologies is based on generating candidate regions, which is divided into one-stage detection algorithms and two-stage detection algorithms. Girshick proposed Fast RCNN [6], which employs VGG16 [7] as the feature extraction network. The border regression process is directly incorporated into the CNN network training, and Softmax is utilized for target classification. The testing and training speeds have been significantly enhanced. However, the algorithm continues to utilize a selective search for identifying potential regions, a process that is notably time-consuming. Ren proposed the Faster RCNN algorithm [8] as an improvement on Fast RCNN. It uses the Region Proposal Network (RPN) to generate candidate regions instead of selective search, which greatly reduces the detection time and improves the accuracy. In 2017, Lin proposed feature pyramid networks (FPNs) based on Faster RCNN [9]. This network transmits semantic features through a top-down structure and achieves feature fusion at different scales through horizontal connections, thereby effectively using the semantic information of low-resolution images. This enables feature maps at different scales to contain strong semantic information.

The two-stage object detection network is capable of achieving satisfactory detection accuracy, although it is relatively slow and, therefore, unable to meet the requirements of real-time detection. In 2016, Redmon introduced YOLOv1 [10], effectively eliminating the need for the region proposal stage. The image is divided into several grids, and the bounding boxes and category probabilities of the objects in the grids can be detected. This greatly improves the detection speed compared to the two-stage detection algorithm. Lin proposed RetinaNet [11], which introduced focal loss to address the imbalance between positive and negative samples as well as the imbalance between easy and difficult samples, based on the category imbalance problem during training. Redmon once more proposed modifications to the YOLO algorithm [12]. The YOLOv3 model employs the Darknet-53 network as the feature extraction component and incorporates additional data enhancement techniques. The Ultralytics team proposed YOLOv5, which employs the GIOU bounding box loss function and optimizes the model size. This resulted in the development of multiple model sizes that can be utilized in a range of scenarios. PP-YOLOE [13], proposed by Xu, employs task alignment learning (TAL), a method of classification and positioning proposed by TOOD alignment learning (TAL), which unifies the optimal anchors for classification and positioning and enhances the T-head to obtain the ET-head. The varifocal loss (VFL) of VarifocalNet is employed for classification loss, while the distribution focal loss (DFL) is utilized as the loss for the positioning task. This has led to a notable enhancement in accuracy.

### 2.2. Object Detection Algorithm for Aerial Images

The characteristics inherent to aerial photography of small objects are low resolution and severe occlusion, which impede the model’s capacity to learn and detect features. Consequently, numerous scholars have directed their attention to this issue and conducted research to address the challenges posed by these characteristics.

Zhu proposed TPH-YOLOv5 [14], an improved version of YOLOv5, which was developed by the same research group. The addition of a detection head enables the system to identify objects of smaller scales. The Transformer Prediction Heads (TPH) are employed as the detection head, replacing the original detection head. Nevertheless, the model still exhibits considerable parameter and computational complexity, rendering it unsuitable for deployment on resource-constrained detection devices. DroneNet [15] proposes a feature information enhancement module (FIEM) that can effectively retain target information and can be seamlessly integrated into the backbone network as a plug-and-play module. Moreover, the proposal incorporates a split-connected feature pyramid network (SCFPN). This network not only integrates feature information across various scales but also thoroughly examines feature layers that contain numerous small objects, enhancing its comprehensiveness in feature exploration. The effective utilization of spatial and semantic information is crucial for enhancing the performance of small target detection. Hu proposed the EL-YOLO [16] model and designed an improved feature extraction method, ESPP (Enhanced Spatial Pyramid Pooling), to replace the original SPP (Spatial Pyramid Pooling) method. This improvement markedly enhances the model’s capacity to extract features from small objects. Furthermore, the EL-YOLO model employs α-CIoU (α-Complete Intersection over Union) as the loss function for object detection. The introduction of the parameter α enables α-CIoU to effectively optimize the imbalance between positive and negative samples in the bounding box regression task. Despite the deployment of the YOLOv5s model, there remains room for further enhancement. In order to better adapt to the deployment requirements of edge devices, it would be beneficial to consider the YOLOv5n model, which is more lightweight, with a view to further reduce the number of parameters and computational costs. Tao [17] proposed YOLOv5s_2E, introduced Soft_NMS [18], and combined it with EIoU to propose the EIoU_Soft_NMS algorithm, which replaced the non-maximum suppression algorithm of the original network to enhance the detection of occluded objects. Introducing Focal EIOU [19] into the boundary box regression loss function facilitated accelerated network convergence, reduced loss, and enhanced system inference accuracy. The Focal EIoU loss function is designed to prioritize the training of high-quality samples. However, in aerial image datasets, the majority of samples are challenging samples with minimal IoU overlap. This makes Focal EIoU particularly ineffective in this context. The Focal EIoU loss function is not sufficient to handle difficult samples, which may affect the detection and positioning accuracy of small objects and, in turn, reduce the overall detection performance of the model. This model also integrates the attention mechanism with the detector head by combining it with the dynamic head framework DyHead [20], thereby enhancing the accuracy of small object detection. Muzammul proposed an aerial image analysis method [21] by combining the real-time detection and recognition (RT-DETR-X) model [22] with the slice-assisted hyper-inference (SAHI) method [23]. Despite the RT-DETR model being an end-to-end detector without the pre-processing of anchor boxes and post-processing of NMS, which simplifies the detection process and reduces the potential propagation of errors, the SAHI method further enhances the model’s reasoning ability. The slicing processing technology enables the model to perform fine detection at higher resolutions, thereby capturing more detailed information. Nevertheless, the model still comprises a considerable number of parameters and calculations, and there is still scope for further optimization.

Furthermore, the detection equipment utilized by resource-constrained UAVs necessitates specific model parameters and computing capabilities. Scholars have also conducted pertinent research in this field, striving to minimize the model’s size.

Wang proposed YOLO-ERF [24], which designed a lightweight backbone network by introducing the ERF module to expand the receptive field of the network. The ERF module optimized the path aggregation network structure and drastically cut down the number of network parameters while also enhancing the network’s capability to detect small objects within complex backgrounds. Furthermore, Wang proposed the implementation of a lightweight detection head, which would serve to enhance the accuracy of the model’s detection capabilities. This approach optimizes the path aggregation network structure while maintaining a lightweight design for the model, thereby reducing computational complexity and parameters. Although YOLO-ERF has made significant progress in terms of parameters and detection performance, the high memory access cost and frequent memory access times caused by the multi-branch structure network may still be a bottleneck for network performance. These issues may result in a reduction in the speed of inference in practical applications, which could impact the real-time performance and efficiency of the model. Cao put forth a proposal for a lightweight YOLO network (GCL-YOLO) [25]. Firstly, a backbone network based on GhostConv [26] was constructed, which was capable of reducing the number of parameters by half without any loss of accuracy in object detection. The feature fusion network in the YOLOv5 network was reconstructed. A shallow structure fusion layer and a novel object prediction head were incorporated to ensure the precise localization of high-density, small objects. Concurrently, the prediction head for large experimental objects was removed in order to further reduce the number of network parameters. Finally, the Focal EIOU loss function is employed as the bounding box loss function. Similarly, the objective of this method is to direct greater attention to high-quality samples while simultaneously minimizing the processing of challenging samples. Wang proposed MFP-YOLO [27], which incorporates a multi-path inverse residual module that is coupled with an attention mechanism. This mechanism is designed to address issues related to significant scale changes and rich interference in complex backgrounds. Additionally, a lightweight decoupled head has been developed, which replaces the original model’s detection head. This modification has the effect of accelerating the network’s convergence speed and reducing the number of parameters in the original model’s detection head.

## 3. Materials and Methods

### 3.1. Proposed Model

#### 3.1.1. Model Architecture

The YOLO series of detection algorithms is recognized for its remarkable efficiency and precision in real-time object identification. It is a widely used tool in a variety of object detection tasks. YOLOv8 represents the latest iteration of this series, enhancing the architecture and performance of the model. The YOLOv8n model represents the most compact variant within the YOLOv8 series, designed for deployment in scenarios characterized by extreme latency constraints and resource limitations. It has a minimal number of parameters and calculations, rendering it suitable for execution on embedded and mobile devices while also constraining its detection performance when confronted with complex samples. The efficient architecture of YOLOv8n enables real-time object detection on low-end hardware. Aiming at small object detection tasks in order to overcome the existing shortcomings and enhance the detection performance of the model while reducing the number of parameters and the number of calculations to a minimum, we propose the ESOD-YOLO (Efficient Small Object Detection YOLO) detection algorithm, as shown in Figure 2. The algorithmic structure is designed to address the challenge of dense small object recognition in aerial images.

The ESOD-YOLO model is based on the YOLOv8n algorithm. The structure of YOLOv8n comprises three distinct parts: the backbone, neck, and head, which are the three main components of the model. The RepNIBMS module is proposed as a replacement for the C2f module of the YOLOv8n feature extraction network. Its purpose is to enhance the model’s ability to process objects of different scales and to fuse features of different levels in a hierarchical manner. The neck part of the structure has been modified to incorporate a multi-scale feature fusion structure across multiple layers. This integration of semantic and spatial information from feature maps of different scales at the current layer, the upper layer, and the lower layer are intended to enhance the model’s feature expression ability following fusion.

#### 3.1.2. RepNIBMS Module

The scale of the target in the aerial image is heterogeneous. As shown in Figure 3, the dimensions of the various objects exhibit a considerable degree of variability. Furthermore, the scale of an object in an image may vary depending on the shooting angle and distance, even if the object itself remains constant. It is therefore evident that the processing of multi-scale features plays an integral role in the training and detection of models. The manner in which multi-scale features are processed, both in terms of simplicity and efficiency, has a profound impact on the overall efficiency and accuracy of the algorithm.

Figure 4a demonstrates the CSP feature extraction module utilized by YOLOv5. The primary aim of this structure’s design is to minimize computational load by segregating feature maps across various stages and tackling the problem of redundant gradient information. Consequently, it enhances both the accuracy and speed of the model. However, this structure has limited flexibility and does not optimally utilize gradient flow information. Therefore, the YOLOv8 proposes the C2f module, as illustrated in Figure 4b. The module simplifies the process of feature fusion and reduces the number of redundant computations. A more efficient method is employed for channel separation and fusion, which serves to reduce the computational complexity and improve the overall efficiency of the model. Furthermore, the structure optimizes the gradient flow path to ensure the flow of gradients in the deep network, thereby enhancing the stability of training.

While this module in the YOLO series demonstrates commendable performance in general object detection tasks, it exhibits certain limitations when applied to specific contexts, such as aerial imagery captured by drones. Notably, challenges related to small object detection within aerial images, complex background processing, and constraints on computational resources necessitate more specialized solutions.

However, the majority of existing YOLO algorithms merely perform feature fusion in the neck structure to obtain comprehensive multi-scale semantic information while failing to acknowledge the significance of feature extraction networks in constructing multi-scale feature representations. Thus, it is imperative to enhance the module to guarantee its capacity to learn more nuanced multi-scale information. Concurrently, a module, RepNIBMS, is devised by fusing the concepts of the hierarchical aggregation network DLA [28] and Res2Net [29], which enables the aggregation of hierarchical residuals. This structure has the capacity to enhance the number of scales present in the output features, facilitate the learning of more expressive multi-scale features, and maintain a rapid calculation speed. Specifically, the RepNIBMS module first partitions the input data into multiple subsets, each of which is processed independently by a RepIB module. The RepIB module enables the aggregation of residuals across different levels, thereby enabling the model to acquire richer feature representations and achieve deep feature aggregation alongside efficient processing. This methodology effectively captures feature information at various scales, thus enhancing the learning of more expressive multi-scale features. Simultaneously, RepNIBMS ensures rapid computational speed, rendering it suitable for real-time applications.

Figure 4c demonstrates the RepNIBMS module. Let *X* be an input feature represented by X ∈ R^H×W×C^ followed by a 1 × 1 convolution. The channel dimension of X is expanded to n × C, and X is divided into n different groups, denoted as {*X_i_*} *I* ∈ 1, 2, 3, …, *n*. To reduce the computational cost, we take *n* = 3. With the exception of the initial group, each subsequent group is required to undergo the RepIB module, which is represented by RepIB_3×3_(·). The output, denoted by *Y_i_*, is expressed mathematically in Equation (1):(1)Yi =XiRepIBk×k(Yi−1+Xi)  i=1i>1

The reparameterized inverted block (RepIB) module is employed in accordance with the aforementioned formula to facilitate cross-layer connection, thereby ensuring a more seamless flow of gradients within the deep network. Finally, all the hierarchical items are connected, and the information is exchanged through 1 × 1 convolution, which enhances the model’s object detection ability in complex scenes. The RepIB structure, as defined in Equation (1), is shown in Figure 5. An increase in the network’s depth and width can lead to accuracy improvements. To this end, a module with an inverse residual has been designed [30,31]. Firstly, the number of channels is expanded through a 1 × 1 convolution, with an expansion factor of 4. Secondly, the RepConv structure is employed, with a multi-branch structure model used for training in order to optimize the performance of the network. During the inference phase, the concept of structural reparameterization [32] is employed to transform the network into a single-channel structure, which has the effect of reducing both memory usage and running speed. Subsequently, the network is downsampled by a 1 × 1 convolutional layer and passes through a dropout layer. During the training phase, a random selection of neurons is discarded, resulting in a different network structure at each training stage. This prevents the network from overfitting the training data, thereby improving the generalization ability and robustness of the model. The dropout rate is set to 0.1.

#### 3.1.3. WFPN Module

The feature information of multiple scales is extracted in the feature extraction network. Subsequently, the extracted features require further processing and fusion. As illustrated in Figure 6b, the low-level feature map is generated in the initial layer of the network and comprises a substantial quantity of spatial details and edge information as well as the majority of the original information present in the input image. As demonstrated in Figure 6c, the high-level feature map is generated in the deep layer of the network. It contains more semantic information, with less spatial detail, and is capable of extracting global semantic information. The combination of these feature maps of varying scales and levels results in the generation of more nuanced and discriminating features.

The YOLO series of algorithms employs the feature pyramid network (FPN) [9] and path aggregation network (PAN) [33] structures, as illustrated in Figure 7a,b. These structures permit the network to utilize features of varying scales, thereby facilitating the acquisition of contextual information and enhancing the algorithm’s capacity to detect both small and large objects. However, the flow path of contextual information in this structure is fixed and not sufficiently flexible. Consequently, a cross-level multi-scale feature fusion structure, WFPN, is proposed.

As shown in Figure 7c, similar to waves advancing from the deep sea to the shallows, each layer of these waves carries energy at varying depths and speeds, intermingling and intertwining to create a complex and orderly dynamic system. In the WFPN structure, feature information is propagated in an analogous manner. Rather than flowing in a single direction, it surges across multiple scales simultaneously; each feature map functions like a wave that receives vertical ‘surge’ information from the preceding layer while also ‘colliding’ and ‘fusing’ with horizontally connected peers at the same level. This structure demonstrates superior performance in the processing of multi-scale features. The incorporation of additional scale features and more sophisticated fusion strategies enables the more precise processing of objects across a wider range of scales, thereby enhancing the model’s capacity to detect objects of varying sizes. At the same time, the complexity of this structure is lower than that of other feature fusion methods.

In the structure of the neck part, the feature information of different scales should be fully combined. However, the fusion mechanism of Generalized-FPN (GFPN) [34] will lead to an increase in the number of parameters and calculations. An increase in the number of complex feature fusions and processing will inevitably lead to an increase in resource consumption. Accordingly, we devised a WFPN configuration akin to a wave, which both establishes horizontal connections between feature maps at each level and vertical connections from the previous level, and it also accepts inputs from the previous level of horizontal connections. This structure is not constrained by a fixed flow path of contextual information, and the introduction of multi-path connections allows connections to be freely established between feature maps of different scales. Each feature map can benefit from information at multiple scales. Low-level feature maps (containing detailed information) and high-level feature maps (containing semantic information) complement each other, thereby enhancing the overall feature expression ability. The model connections are simple and efficient, ensuring multi-level feature fusion. The data flow of Figure 2 is drawn with reference to the structure of Figure 7c, as shown in Figure 8. We utilized the position features of the WFPN structure in the network model multiple times to enhance the utilization of detailed information.

This connection method can be expressed by the following equation: For the kth column, *k* ∈ 1, 2, 3, …, n, the feature map of the lth layer, *l* ∈ 1, 2, 3, …, n, will accept the features of the previous layers. In the case where k is equal to zero, this indicates that the feature extraction network backbone is being considered.(2)Pkl=Conv(Concat(Pkl−1,Pk−1l,Pk−1l−1))Conv(Concat(Pkl+1,Pk−1l,Pk−1l−1))  k=2nk=2n+1

The function *concat(·)* represents the concatenation of the feature maps generated in all previous layers. The function *conv(·)* represents a 3 × 3 convolution. The structure can be expanded indefinitely, with k set to 2 to form the network structure depicted in Figure 2.

#### 3.1.4. Additional Detection Head for Small Objects

Aerial images often contain a large number of small-scale objects, which makes detection significantly more difficult due to their small size in the image. When processing these images, traditional feature extraction networks usually reduce the size of the feature map by downsampling multiple times to extract more semantic information. However, this downsampling process inevitably loses a lot of detailed spatial information. The provision of detailed information is important for the detection of small-scale objects. In the neck part of the deep learning model, upsampling operations are employed to restore the resolution of the feature map, thereby enabling the fusion of information from different levels. Nevertheless, even with these upsampling operations, the original feature map underwent a significant loss of key information during downsampling. The reliance on traditional three-detection-head processing for this multi-scale information often results in ineffective small-scale object detection, which can lead to missed detections. In order to effectively tackle this issue, it becomes imperative to incorporate a specialized detection head tailored specifically to the identification of small objects. 

The original YOLOv8n model was equipped with three detection heads, each of which processes feature maps of 20 × 20, 40 × 40, and 80 × 80 sizes in order to detect objects of varying scales. However, this configuration renders the system less effective in detecting objects of a minute scale. To address this limitation, a fourth detection head with a resolution of 160 × 160 was incorporated into the existing three detection heads. The detector was provided with feature maps that have been downsampled four-fold, resulting in the fusion of spatial and semantic information. This approach allows for the retention of more detailed spatial data and the integration of semantic information, enhancing the overall performance of the detector. The small object detection head, with a size of 160 × 160, exhibits enhanced positional detail. Despite the increase in parameters and model calculation, the incorporation of this diminutive object detection head markedly enhances the model’s capacity to discern minute entities, curtails misdetections, elevates overall detection precision, and facilitates more efficacious detection of minute objects. This, in turn, augments the model’s ability to discern objects in intricate scenes comprising a multitude of minute components. The difference can be seen in Figure 9.

#### 3.1.5. Tri-Focal Loss Function

The YOLOv8 algorithm employs a loss function comprising two distinct components: classification loss and regression loss. The classification loss employs the BCE (binary cross-entropy) loss function. As a consequence of the removal of object loss and object confidence in the output by YOLOv8, the confidence scores for each category are directly output, with the maximum value being taken and used as the confidence of the anchor box. The regression loss is formulated as a DFL [13] loss in conjunction with a CIOU loss. The DFL loss function is designed to optimize the probability of the two positions that are most closely aligned with the label. This allows the network to rapidly focus on the target position and the distribution of neighbouring regions.

As demonstrated in Equation (3), CIOU considers three geometric factors: minimization of the normalized centroid distance, overlap area, and consistency of the aspect ratio.(3)$$L_{CIOU}=1-IOU+\frac{\rho^{2}\left(b,b^{gt}\right)}{c^{2}}+\alpha \nu \;\nu =\frac{4}{\pi^{2}}(arctan\frac{w^{gt}}{h^{gt}}-arctan{\frac{w}{h})}^{2}\;\alpha =\frac{\nu }{(1-IOU)+\nu }$$
where *b* and *b^gt^* represent the centre points of the prediction box and the ground truth box, respectively. However, in the case of aerial image object detection, the dataset comprises a significant proportion of challenging samples and a relatively small number of simple samples. It is generally accepted that samples presenting greater difficulty are more challenging to detect. If the model is able to more effectively process these samples, the overall detection performance will be markedly enhanced. Consequently, it is essential to identify challenging samples and assign them a greater significance within the loss function, thereby prompting the model to prioritize the regression of these challenging samples during training. This paper proposes a simple and effective weighted loss function that enhances the model’s focus on high-quality examples based on the original border loss. Concurrently, in order to minimize the number of hyperparameters, we utilized *IOUT* as the sole adjustable hyperparameter of this method, which is categorized into three intervals. Each interval is assigned a distinct weight of attention to samples, with the objective of allocating a higher weight to challenging samples. For samples that are easily classified, we assigned them a reduced weight to minimize their contribution to the loss function. For those of medium difficulty, we allocated a moderate weight to ensure that the model effectively learns their features during training. In contrast, for difficult samples, we assigned a greater weight to encourage the model’s increased focus on these challenging instances throughout the training process.

As illustrated in Equation (4), *L*_Tri-focal CIOU_ is employed as a substitute for L_CIOU_:(4)$$L_{Tri-focalCIOU}\;=W_{Tri-focal}L_{CIOU}$$

*W*_Tri-focal_ represents a weighted variant of the original loss function, as shown in Equation (5):(5)$$W_{Tri-focal}=\left\{\begin{array}{ll} e & IOU\leq 1-IOUT\\ e^{1-IOUT} & 1-IOUT<IOU<IOUT\\ e^{1-IOU} & IOU\geq IOUT \end{array}\right.$$

Figure 10 illustrates the graph of the hyperparameter *IOUT*, which was set to 0.4. When the *IOU* is low, it can be inferred that the sample is challenging at this point in time. The border loss, multiplied by this weight, has the effect of concentrating the loss on difficult examples.

## 4. Experiments

### 4.1. Experimental Setup

#### 4.1.1. Datasets and Evaluation Metrics

In order to ascertain the efficacy of the proposed model, experiments were conducted on the public aerial image benchmark dataset VisDrone [2]. The dataset comprises 10,209 static images captured by the drone camera. Of these, 6471 images were utilized for training purposes, 548 images for verification, and 1580 images for testing. The dataset comprises a diverse range of scenes, including urban and rural settings, highways, public squares, and stadiums, encompassing a multitude of intricate environments. At the same time, the backgrounds of the images are complex and variable, including different weather, lighting conditions, and occlusion, which enhances the robustness of the model in practical applications. The dataset contains a wide variety of target categories, including pedestrians, vehicles, bicycles, trucks, buses, motorcycles, tricycles, etc. This diversity makes the dataset highly representative and practical for real-world applications. We use VisDrone2019-DET-train as the training set and VisDrone2019-DET-test-dev as the test set. Figure 11 gives a visual example of these challenges.

The evaluation metrics of the experiment are precision, recall, mAP@0.5, mAP@0.5:0.95, GFLOPs, and parameters. Precision indicates how many of the positive samples are correct, recall indicates how many of the positive samples are correctly predicted, the average precision AP indicates the area under the P-R curve obtained by taking precision as the vertical axis and recall as the horizontal axis, and mAP indicates the average precision of each category. M AP@0.5 indicates the average detection accuracy when the IOU threshold is set to 0.5 for each category, and mAP@0.5:0.95 is the average detection accuracy calculated at a step size of 0.05 for all IOU thresholds between 0.5 and 0.95. GFLOPs are used to measure the speed of the model, and parameters are used to measure the size of the model.

#### 4.1.2. Experimental Environment

The software environment and hardware parameters are presented in Table 1.

The parameters used for training ESOD-YOLO in this experiment are presented in Table 2.

### 4.2. Experimental Results

#### 4.2.1. Ablation Experiment

In order to facilitate a more comprehensive analysis of the influence of each improvement point in the proposed ESOD-YOLO algorithm on the algorithmic performance, we present a visual demonstration of the results through ablation experiments. 

In this illustration, A represents the proposed WFPN module, B represents the additional small object detection head, C represents the RepNIBMS module, and D represents the proposed tri-focal CIOU loss function.

As evidenced in Table 3, each of the model schemes demonstrates the capacity to enhance the precision of object detection.

The WFPN module enhances the model’s detection accuracy while maintaining a relatively low increase in parameters and computational load. The incorporation of a compact object detection head markedly enhances the precision of the model’s detection capabilities. However, it also necessitates the introduction of supplementary feature maps for detection, consequently leading to more intricate feature fusion and processing. These steps result in a notable increase in the number of parameters and the number of calculations required by the model. The RepNIBMS module enhances the precision of the model while concurrently reducing the number of parameters and calculations associated with the model. Furthermore, the incorporation of the tri-focal function has been demonstrated to enhance the accuracy of the model.

The final model exhibits an mAP@0.5 increase of 3.7%, an mAP@0.5:0.95 increase of 2.4%, a 1.46 M increase in the number of parameters, and a 6.2 G increase in the number of calculations.

#### 4.2.2. Comparative Experiment of RepNIBMS Module

In order to ascertain the efficacy of the RepNIBMS module, a comparison was undertaken with other modules in the YOLO series. C2f is derived from the YOLOv8 model, the MSBlock is derived from the YOLO-MS model [35], and RepNCSPELAN4 is derived from the YOLOv9 model [36]. The results are presented in Table 4.

As can be seen from Table 4, the RepNCSPELAN4 [36] module has the fewest parameters and the least number of calculations, but the detection effect after incorporating this module is also the worst. The RepNIBMS module proposed in this paper not only improves the detection accuracy based on YOLOv8n but also slightly reduces the number of parameters and the number of calculations.

#### 4.2.3. Comparative Experiment of WFPN Module with Additional Detection Head

In order to verify the effectiveness of the WFPN module, a comparison was made with other feature fusion modules. The incorporation of a small object detection head necessitated the introduction of supplementary feature maps for detection purposes. An additional layer was incorporated into the neck section for the purpose of feature fusion. The modules presented here are all situated subsequent to the addition of the small object detection head. The FPN + PAN module was derived from the YOLOv8 model, and the BiFPN module was derived from the EfficientDet model [37]. In the context of comparative experiments utilizing BiFPN, it is essential to establish the dimensions of feature maps. While a scale of 128 was chosen to balance computational complexity and accuracy, this setting resulted in significantly high computational demands that were ill-suited for resource-constrained environments such as drones. The results are presented in Table 5.

Table 5 demonstrates that WFPN is capable of achieving superior detection accuracy in comparison to other modules. However, this enhanced performance is accompanied by a slight increase in the number of parameters and calculations.

#### 4.2.4. Comparative Experiment of Tri-Focal Loss

To verify the effectiveness of the tri-focal function, we added the tri-focal function to CIOU, EIOU, and SIOU separately, and the results are shown in Table 6.

Table 6 demonstrates that the introduction of the tri-focal loss function resulted in an increase in the values of CIOU, EIOU, mAP@0.5/%, and mAP@0.5:0.95/%. Furthermore, the value of SIOU mAP@0.5/% also increased. This indicates that the original model became more attentive to the challenging samples, thereby enhancing the detection efficacy.

The efficacy of the tri-focal weighted loss function was assessed through the examination of the precision and recall curves, as illustrated in Figure 12, which shows that the accuracy and recall of the model exhibited slight improvement following the incorporation of the tri-focal weighted loss function. The model demonstrates an enhanced capacity to identify true positive samples while concurrently reducing false positives. This suggests that the novel loss function is more efficacious in processing all samples, particularly the challenging ones, and thereby, it can markedly enhance the overall performance of the model.

#### 4.2.5. Comparative Experiment of Different Models

In order to ascertain the efficacy of the model, a number of widely used models were selected for comparison. RetinaNet is a classic two-stage detection algorithm with high detection accuracy and can be used as a high-precision benchmark model to measure the performance of the proposed model in terms of accuracy. Lightweight models such as YOLOv3-tiny and YOLOX-tiny can reflect the improvement effects of the proposed model in lightweight design. Different scales of the YOLOv5 version help to show the advantages of the proposed model in dealing with the small object detection problem of aerial images compared with the YOLOv5 series. Comparison with YOLOv8n as the basic model can intuitively present the improved effect. Comparison with the larger scale and better performing YOLOv8s can highlight the advantages of the proposed model in maintaining high detection accuracy while having advantages in parameter and calculation numbers. The accuracy, number of parameters, and computational effort of each model were then evaluated. The training parameters for all the compared models were the same as those of ESOD-YOLO shown in Table 1. The results are presented in Table 7.

Table 7 illustrates that, with the exception of YOLOv8s, ESOD-YOLO exhibited the highest mAP@0.5/%. Although the mAP@0.5:0.95/% was slightly inferior to that of RetinaNet [11], the parameters and computational load of this model were considerably lower than that of RetinaNet with a similar accuracy. It makes an ideal choice for drones with limited resources. In comparison with alternative lightweight models, the proposed model is demonstrated to have the most effective detection capabilities. The parameters and computational cost of ESOD-YOLO are only half those of YOLOv8s, yet the effect is only slightly inferior to that of YOLOv8s. As interpreted in Table 7, the mAP@0.5 and mAP@0.5:0.95 of the final model are only 1% inferior to YOLOv8s, while the number of parameters and the number of calculations are only half of YOLOv8s. Figure 13 demonstrates that the enhanced model is capable of learning more valuable information.

In order to assess the capacity of the ESOD-YOLO model for identifying minute objects, a comparison of precision was conducted between the original model and the model employed in this study across the ten categories in the VisDrone dataset. The results are presented in Table 8. The metric employed was the average precision (AP) value (IOU = 0.5) expressed as a percentage for each category.

Table 8 shows that pedestrians, people, cars, and motors had the greatest improvement. These categories had the characteristics of a large proportion of small objects, dense distribution, and obvious scale changes, which shows that this model can detect these types of objects well. It can solve more missed and false detection phenomena.

Figure 14a represents the original image, Figure 14b represents the detection result of YOLOv8n, and Figure 14c represents the detection effect of ESOD-YOLO proposed in this paper. Figure 14a depicts the scenario in which two motors are situated in the upper right corner. Figure 14b illustrates a scenario in which one motor is not detected, but the other is identified as a pedestrian. Figure 14c depicts a scenario in which both motors are successfully detected.

Figure 15a depicts the original image, Figure 15b depicts the detection outcome of YOLOv8n, and Figure 15c depicts the detection outcome of ESOD-YOLO as proposed in this paper. Figure 15a depicts a person in the upper left corner. However, the detection result in Figure 15b fails to identify this individual, whereas the proposed method in Figure 15c successfully detects it. Furthermore, the confidence level of the other individuals in Figure 15c is higher than that in Figure 15b, indicating that the model considers the detection result to be correct with a higher probability. This suggests that the model has a higher degree of feature matching for this object.

Figure 16a depicts the original image, Figure 16b illustrates the detection outcome of the YOLOv8n model, and Figure 16c depicts the detection outcome of the ESOD-YOLO model proposed in this paper. In Figure 16a, a group of people can be seen in the upper right corner. The detection result in Figure 16b demonstrates that the YOLOv8n model is unable to detect the group of people and erroneously identifies the bench in the centre of the image as a bicycle. Figure 16c demonstrates that the ESOD-YOLO model is capable of accurately detecting the group of people in the upper right corner of the image without misidentifying the bench in the middle of the image. It demonstrates that the ESOD-YOLO model is more accurate and robust in dealing with small objects and complex scenes and is significantly more effective than the YOLOv8n model.

## 5. Conclusions

In light of the challenges posed by the diverse scales, intricate backgrounds, severe occlusions, diminutive targets, and concentrated distribution of aerial images captured by drones, we put forth an enhanced object detection algorithm, ESOD-YOLO, which is based on the YOLOv8n model. In comparison with the benchmark model, this model demonstrated a notable enhancement in performance across a series of metrics. In the mAP@0.5 metric, the ESOD-YOLO exhibited an improvement of 3.7%, while in the more rigorous mAP@0.5:0.95 metric, it demonstrated a 2.4% enhancement. In the VisDrone dataset, the ESOD-YOLO demonstrated an enhanced capacity to accurately detect the ten designated categories. The gains in accuracy across different categories were as follows. For pedestrians, ESOD-YOLO showed an increase of 7.7 compared to YOLOv8n. In the case of people, there was a gain of 6.7. For bicycles, the gain was 1.9. ESOD-YOLO had a gain of 6.4 for cars. For vans, the gain was 3. For trucks, it was 1. For tricycles, the gain was 4.3. For awning-tricycles, the gain was 0.3. For buses, the gain was 0.1. And for motors, the gain was 5.9. These gains clearly demonstrate the enhanced detection capabilities of ESOD-YOLO over YOLOv8n for different object categories within the VisDrone dataset. These data indicate that the model can adapt to the differences in shape, size, and other aspects of different types of transportation means and has excellent generalization ability.

Despite the model exhibiting an increase in terms of parameters and calculations, the ESOD-YOLO had a notable enhancement in detection accuracy, which renders it a viable option for deployment on edge devices for drones. The model offers a broader scope for potential applications and technical assistance for the advancement of drone-based object detection technology.

Furthermore, considering the requirements and limitations of the model in practical applications, we have formulated a future work plan. One of the important directions is to focus on developing a lightweight design of the ESOD-YOLO model. This design aims to reduce the complexity of the model from multiple dimensions, including but not limited to reducing the number of parameters and optimizing the algorithm’s structure to decrease the number of calculations. Through these measures, it is expected to improve the real-time performance of the model on drone edge devices and enable it to operate more efficiently. While pursuing the improvement in real-time performance, we will also focus on maintaining the accuracy of the model. For this purpose, we will conduct systematic experiments and analyses to study the relationship between both the number of parameters and calculation requirements and the detection accuracy, and we will explore how to ensure that the detection accuracy of the model is not significantly affected when reducing the number of parameters and calculation requirements. Finally, in order to comprehensively verify the practical use value and application effect of the model, we will conduct actual measurements on actual drone edge devices to obtain performance data for the model in a real environment, including but not limited to inference time, detection accuracy, memory occupation, etc., and use these to evaluate whether the model can meet the requirements of practical applications.

## Figures and Tables

**Figure 1 sensors-24-07067-f001:**
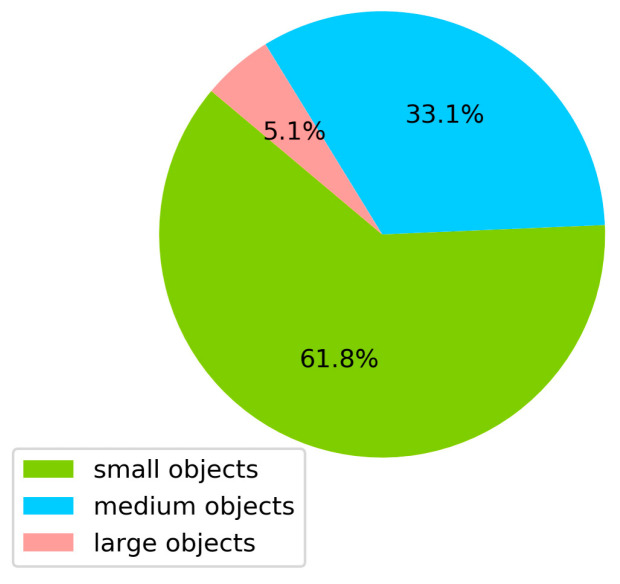
The proportion of large, medium, and small objects in VisDrone dataset.

**Figure 2 sensors-24-07067-f002:**
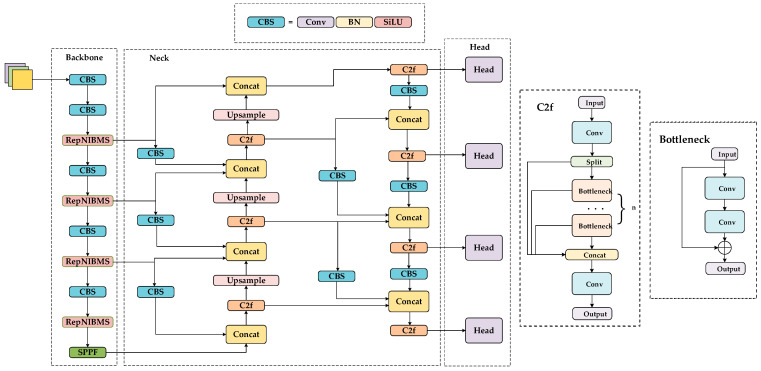
Structure of ESOD-YOLO.

**Figure 3 sensors-24-07067-f003:**
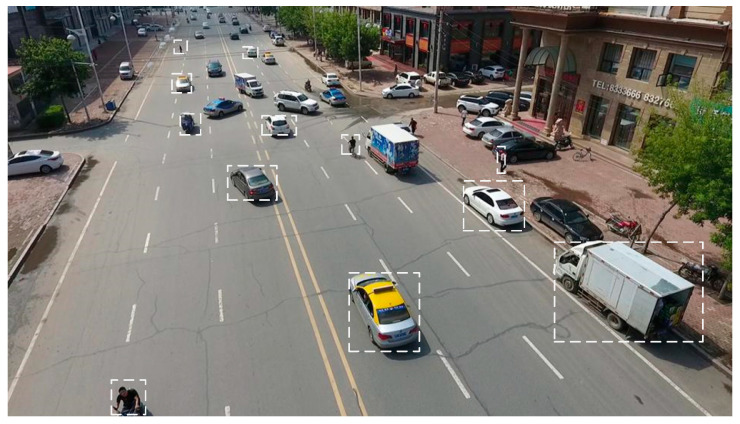
The images were captured from an aerial perspective and depict a range of objects of varying sizes.

**Figure 4 sensors-24-07067-f004:**
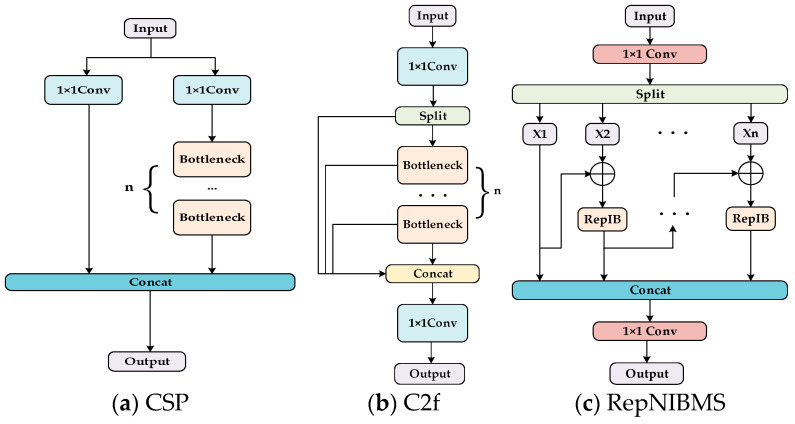
(**a**) Depicts the CSP structure diagram, (**b**) represents the C2f structure diagram, and (**c**) illustrates the RepNIBMS structure diagram.

**Figure 5 sensors-24-07067-f005:**
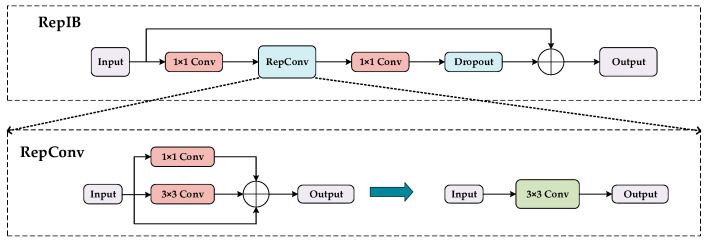
RepIB module structure.

**Figure 6 sensors-24-07067-f006:**
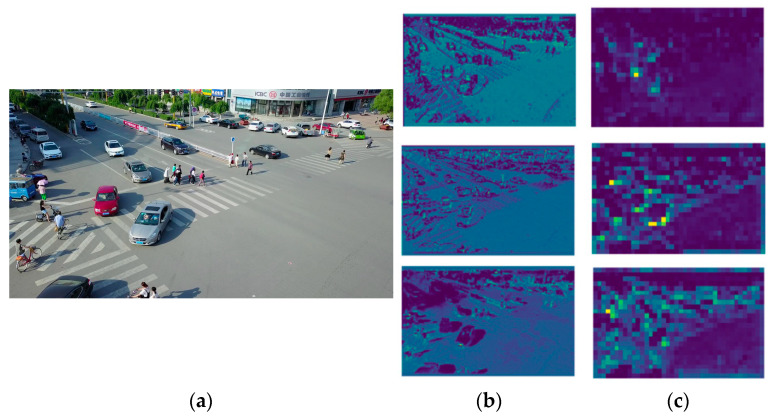
(**a**) Depicts the original input image, (**b**) represents the low-level feature map, and (**c**) illustrates the high-level feature map.

**Figure 7 sensors-24-07067-f007:**
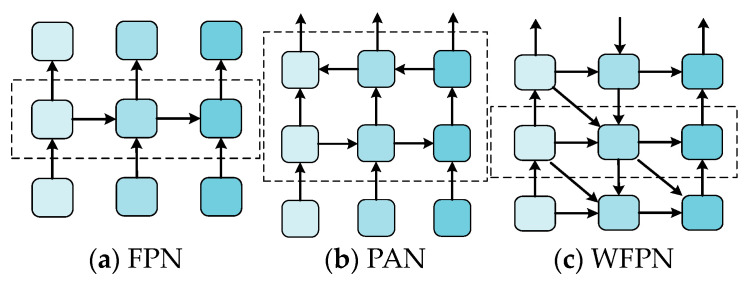
(**a**) Depicts the FPN structure, (**b**) represents the PAN structure, and (**c**) represents the WFPN structure.

**Figure 8 sensors-24-07067-f008:**
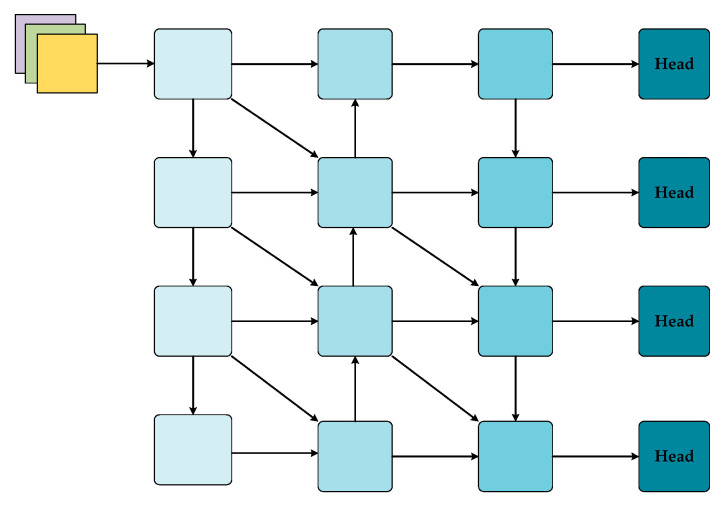
The utilization of the WFPN structure in the network model.

**Figure 9 sensors-24-07067-f009:**
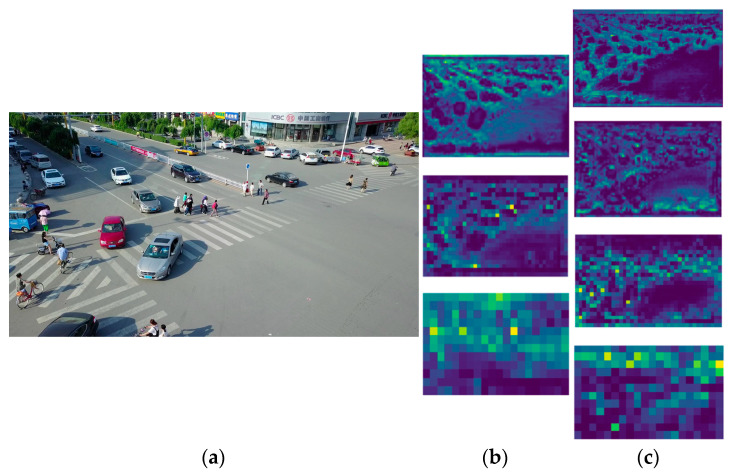
(**a**) Illustrates the image to be detected, (**b**) depicts the feature map of the three output layers without the small object detection head, and (**c**) presents the feature map with the small object detection head.

**Figure 10 sensors-24-07067-f010:**
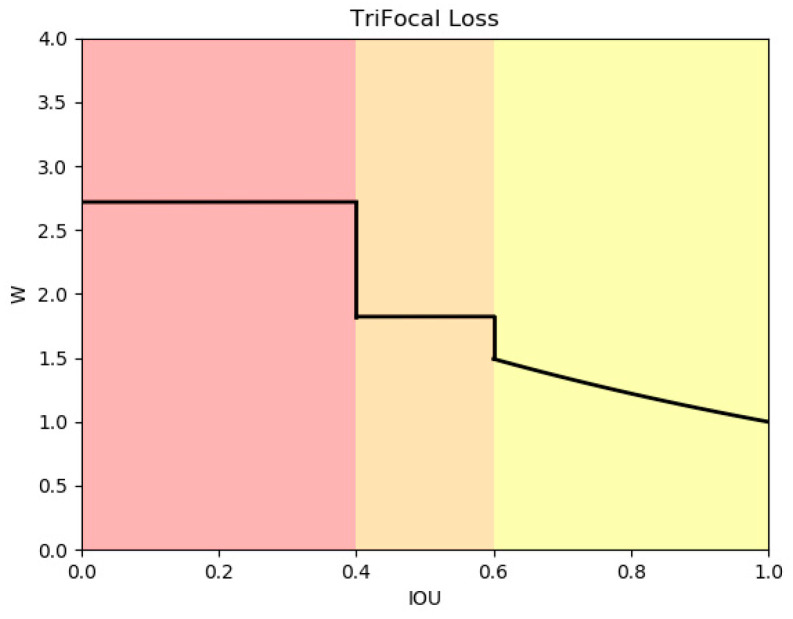
Tri-focal curve diagram with IOU variation. The left side of the chart is shaded in pink, indicating a low IOU range. In contrast, the middle section is orange, representing a medium IOU range, while the right side is yellow, signifying a high IOU range.

**Figure 11 sensors-24-07067-f011:**
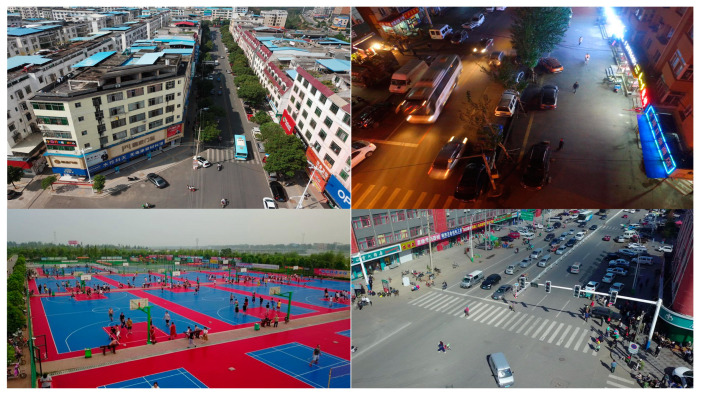
Sample images from the VisDrone dataset.

**Figure 12 sensors-24-07067-f012:**
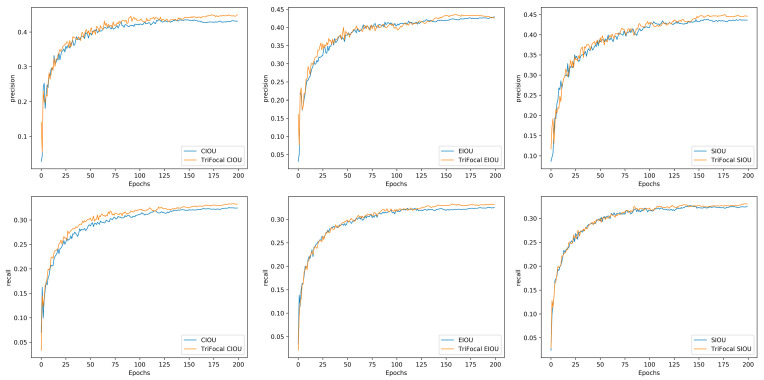
Comparison of the precision and recall curves of the original bounding box regression loss function and the loss function with the tri-focal weighting function.

**Figure 13 sensors-24-07067-f013:**
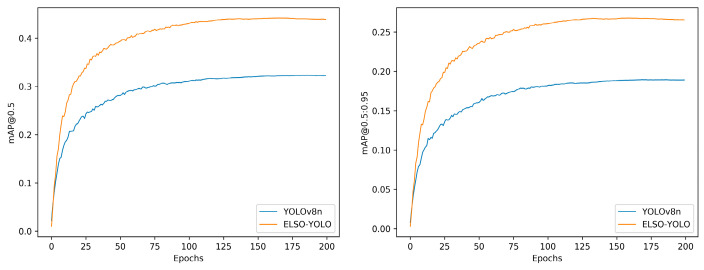
Performance comparison between ESOD-YOLO and YOLOv8n during the training phase.

**Figure 14 sensors-24-07067-f014:**
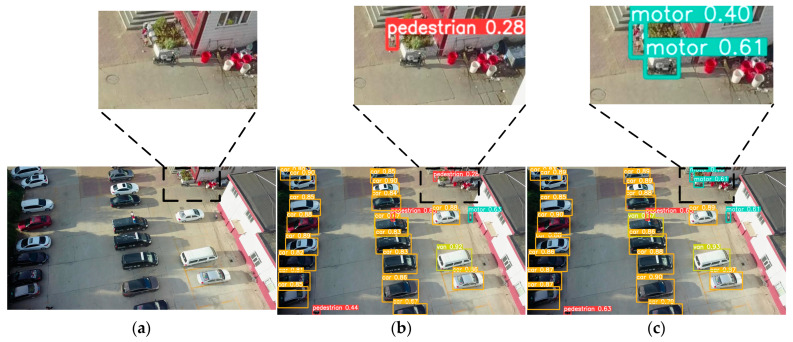
ESOD-YOLO and YOLOv8n’s first visualized detection result. (**a**) Represents the original image. (**b**) Depicts the detection result of YOLOv8n. (**c**) Represents the detection effect of ESOD-YOLO.

**Figure 15 sensors-24-07067-f015:**
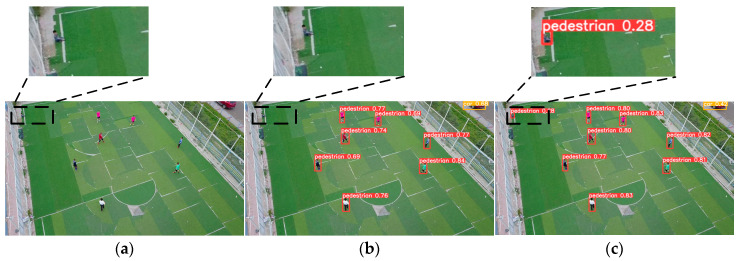
ESOD-YOLO and YOLOv8n’s second visualized detection result. (**a**) Shows the original image. (**b**) Depicts the detection result of YOLOv8n. (**c**) Represents the detection effect of ESOD-YOLO.

**Figure 16 sensors-24-07067-f016:**
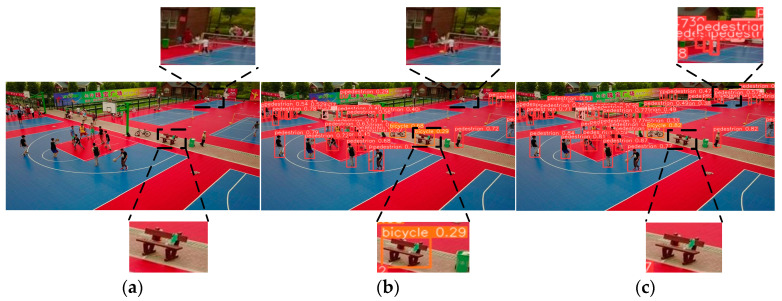
ESOD-YOLO and YOLOv8n’s third visualized detection result. (**a**) Shows the original image. (**b**) Depicts the detection result of YOLOv8n. (**c**) Represents the detection effect of ESOD-YOLO.

**Table 1 sensors-24-07067-t001:** Software environment and hardware parameters.

Platform	Configuration
Integrated development environment	PyCharm 2023.1.2
Scripting language	Python 3.8
Operating system	Ubuntu18.04
CPU	Intel(R) Core(TM) i9-10900K CPU
GPU	RTX 3090
GPU accelerator	CUDA12.2

**Table 2 sensors-24-07067-t002:** ESOD-YOLO training parameters.

Parameter	Configuration
Neural network optimizer	SGD
Learning rate	0.01
Momentum	0.937
Training epochs	200
Batch size	16

**Table 3 sensors-24-07067-t003:** Comparative analysis of ablation.

A	B	C	D	mAP@0.5/%	mAP@0.5:0.95/%	Parameters/M	FLOPs/G
-	-	-	-	25.6	14.2	3.01	8.1
√	-	-	-	26.2	14.5	3.26	8.3
√	√	-	-	28.4	16	4.5	14.3
√	√	√	-	28.9	16.2	4.47	14.3
√	√	√	√	29.3	16.6	4.47	14.3

**Table 4 sensors-24-07067-t004:** Comparative experiment of RepNIBMS module.

Module	mAP@0.5/%	mAP@0.5:0.95/%	Parameters/M	FLOPs/G
YOLOv8n + C2f	25.6	14.2	3.01	8.1
YOLOv8n + MSBlock	25.6	14.1	2.88	7.7
YOLOv8n + RepNCSPELAN4	23.6	13	2.19	5.9
YOLOv8n + RepNIBMS	26.2	14.7	2.97	8

**Table 5 sensors-24-07067-t005:** Comparative experiment of WFPN module with additional detection head.

Module	mAP@0.5/%	mAP@0.5:0.95/%	Parameters/M	FLOPs/G
YOLOv8n + FPN + PAN	27.9	15.6	3.1	12.5
YOLOv8n + BiFPN	28	15.7	2.24	19
YOLOv8n + WFPN	28.4	16	4.5	14.3

**Table 6 sensors-24-07067-t006:** Comparative experiment on tri-focal loss.

Module	mAP@0.5/%	mAP@0.5:0.95/%	Parameters/M	FLOPs/G
YOLOv8n + CIOU	25.6	14.2	3.01	8.1
YOLOv8n + Tri-focal CIOU	26.2	14.5	3.01	8.1
YOLOv8n + EIOU	25.9	14.5	3.01	8.1
YOLOv8n + Tri-focal EIOU	26	14.6	3.01	8.1
YOLOv8n + SIOU	26.4	14.7	3.01	8.1
YOLOv8n + Tri-focal SIOU	26.7	14.7	3.01	8.1

**Table 7 sensors-24-07067-t007:** Comparative experiment of different models.

Model	mAP@0.5/%	mAP@0.5:0.95/%	Parameters/M	FLOPs/G
RetinaNet	29.0	16.9	36.52	210
YOLOv3-tiny	16.2	6.3	61.95	11.7
YOLOX-tiny	23.1	12.4	5.04	3.2
YOLOv5n	23.1	11.5	1.77	4.2
YOLOv5s	26.9	13.8	7.04	15.8
YOLOv8n	25.6	14.2	3.01	8.1
ESOD-YOLO	29.3	16.6	4.47	14.3
YOLOv8s	30.5	17.4	11.13	28.5

**Table 8 sensors-24-07067-t008:** Performance gains of each category in the comparative experiment between the baseline model and the improved model.

Category	YOLOv8n	ESOD-YOLO	Gain
Pedestrian	21.2	28.9	+7.7
People	11.4	18.1	+6.7
Bicycle	5.3	7.2	+1.9
Car	64.4	70.8	+6.4
Van	29.7	32.7	+3
Truck	29.3	30.3	+1
Tricycle	10.5	14.8	+4.3
Awning-tricycle	12.7	13	+0.3
Bus	49.8	49.9	+0.1
Motor	21.8	27.7	+5.9

## Data Availability

Data are contained within the article.

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
