# Peer review of "Efficient Small Object Detection You Only Look Once: A Small Object Detection Algorithm for Aerial Images"

_sensors, 2024, doi:10.3390/s24217067_

Round 1
Reviewer 1 Report
Comments and Suggestions for Authors
line 40: 32*32 unit missed
line 254: acronym “YOLO” is defined here. But it appeared many times before
line 371: GFPN was not defined.
which part of Figure 2 does WFPN(figure 7c) in ?
how does the proposed method related to the existing methods descipted in Sec.2.2.
in Experiments, the comparison with other method for similar problem is necessary,not merely with the classic detection methods.
the introduction and related works can be more concise.
Comments on the Quality of English Languagemany grammer errors, such as
line 42, repeated illustrate
line 44 repeated Figure
line 272 "The structure of YOLOv8n comprises three distinct parts, as follows: The backbone, neck and head are the three main components of the model."
Reviewer 2 Report
Comments and Suggestions for Authors
This manuscript proposes a novel effective small object detection algorithm, called ESOD-YOLO, for aerial images. ESOD-YOLO algorithm is improved based on YOLOv8n. Some limitations of YOLOv8n are improved in this manuscript. The language is concise and readable. Some comments are as follows:
1. The experimental results indicate that for the detection of vehicles, pedestrians, etc., it is recommended to do some detection experiments on plants, animals and hiding targets.
2. The manuscript does not mention why chose the models to do comparative experiment.
3. The parameter settings of comparative algorithms are not point out.
4. The subgraphs of Figure 13 and 14 do not have titles.
5. Some grammar and type errors occurred in the manuscript. E.g. “The parameters for training ESOD-YOLO are shown in Table 1.” Please check the whole manuscript carefully.
6. The best results in the experiment section are not marked.
Comments on the Quality of English LanguageThe quality of English language is fine, and only some type erros occured in the manuscript. Minor editing of the English language is required.
Reviewer 3 Report
Comments and Suggestions for Authors
This manuscript presents an efficient small-object detection model for remote sensing based on YOLO8n. The paper, however, suffers from limitations such as inadequate experimental argumentation and errors in writing format, the specific concerns are detailed below:
1. The innovation of the article is not pronounced. The proposed block-wise residual structure, addition of small object detection layers, multi-scale feature extraction, and fusion are relatively common techniques, and the detection results do not show significant performance improvements compared to existing methods.
2. Using only methods from the YOLO series as comparative algorithms does not substantiate the advanced nature of the algorithm in remote sensing small-object detection. Comparisons should be made with the most advanced methods currently used in remote sensing small-object detection, such as SDMNet,DFFN.
3. Verify whether there is an error in the WFPN in Figure 7c and the Neck structure in Figure 2. The Figure should employ more distinct lines for illustrating data up-sampling and down-sampling to depict the direction of data flow more clearly.
4. Attention should be given to the uniformity of the fonts used in figures and formulas in the paper and to common formatting errors.
5. It is necessary to incorporate an evaluation metric for the model's detection speed.
Comments on the Quality of English LanguageThe English description needs to be revised to make it closer to the native language reading
Reviewer 4 Report
Comments and Suggestions for Authors
The paper presents a novel object detection algorithm for Unmanned Aerial Vehicles (UAVs) based on the YOLOv8n framework, referred to as ESOD-YOLO. Its goal is to address the challenges of detecting small objects in aerial images while reducing the model's complexity for hardware-constrained UAVs. The key contributions include the RepNIBMS module for multi-scale feature representation, the WFPN module for cross-level feature fusion, a small-object detection head, and a Tri-focal loss function to focus on difficult samples. Experimental results show an improvement in detection accuracy on the VisDrone dataset, especially for small objects, while maintaining a low number of parameters and computations suitable for UAV deployment. Despite its promising results, several improvements and clarifications are required.
Major issues:
1. The RepNIBMS module appears highly inspired by Res2Net architecture, which boosts feature representation by combining feature maps with varying receptive fields. The mention of incorporating ideas from Deep Layer Aggregation (DLA) mirrors the approach already taken by the Res2Net framework in terms of multi-scale feature extraction through hierarchical residual connections. It is common in the YOLO series (especially in later versions like YOLOv5 and YOLOv6) to add modules like C3 replacing bottleneck structures in older versions for better gradient flow and multi-scale processing. Hence, the RepNIBMS module is not an outstandingly novel solution but rather a refinement of existing strategies.The practicality of this module is diminished because the multi-scale feature aggregation encapsulated in the original architecture already had reasonable support with models such as YOLOv5 and YOLOv6 , which utilize multi-scale feature branches effectively without the added complexity of hierarchical aggregation networks, such as what RepNIBMS aims for. It would be useful if the authors conducted in-depth comparisons with other scale-aggregation techniques, such as using RepVGG or possibly Dynamic Convolutions to validate the specific improvements. Numerical empirical evidence must support why this added complexity is justified, especially in the context of resource-constrained edge devices for UAVs. How the hierarchical design aids small-object detection better than traditional backbones needs more theoretical and empirical justification.
2. The WFPN is an extension and refinement of familiar structures like FPN and PAN , specifically designed to enhance multi-scale feature processing. Given this, it doesn’t bring a novel concept in terms of the general principle of feature fusion across multiple scales. FPN and PAN already enable flexible exploitation of multi-scale features, helping detect objects of different sizes effectively on deeper and shallower layers. The idea of "wave" connections could be argued to resemble BiFPN (Bidirectional Feature Pyramid Networks) used in the EfficientDet models, which modulate connections and weights dynamically between layers. Moreover, PaFPN (Path-Aggregation Feature Pyramid Network), used in YOLOv5, also uses intricate long-path and short-path fusion mechanisms. Therefore, WFPN is not fundamentally innovative, but an evolution on the same concepts present in feature pyramid architectures. While the "wave-like" structure is mentioned anecdotally, there is insufficient mathematical or algorithmic rigor to clearly delineate how it differs fundamentally from its predecessors like FPN, PAN, or BiFPN, other than the claim of offering an undetermined "superior" performance. The increase in parameters and computational cost due to this elaborate feature fusion mechanism raises concerns, especially since one of the paper’s main objectives is deployment on resource-constrained UAV platforms. The attainable benefit of the wave transmission feature for small objects might not justify this added complexity. More concrete diagrams and clearer formulations are needed to justify how "wave-flow" differs from existing bidirectional methods like BiFPN. Additionally, parameter and computation growth should be carefully analyzed in the context of real-world UAV deployment (e.g., inference time, memory usage, etc.).
3. The model introduces an additional detection head specifically for small objects in the 160x160 resolution range. This dedicated head is meant to increase the effectiveness of detecting tiny targets by fusing more shallow-level features with spatial detail that could be lost in the broader detection heads. This approach is more of an adaptation of the architecture already seen in previous YOLO models, such as YOLOv3 and YOLOv5, which include multiple detection heads for detecting objects at different scales. In earlier models like YOLOv5, three heads handle large, medium, and small objects across varying resolutions. While this fourth head may help in small-object scenarios, it comes at the expense of extra parameters and computational cost. Importantly, the paper fails to perform a convincing trade-off analysis regarding the marginal gain from the added detection head compared to the incurred computational complexity, which is worrisome for UAV-based edge devices. Additionally, this modification—while potentially useful—doesn't systematically address issues like overfitting, balance across detection heads, or hierarchical learning efficiencies. It is an ad-hoc solution that doesn't innovate beyond what existing multiple-head detection techniques offer. A more detailed performance vs. complexity analysis should have been conducted here. Specifically, it would help to see how much additional inference time or memory usage the fourth detection head consumes and if this negatively affects deployment on edge devices. The authors should consider presenting a trade-off study that looks at how introducing the extra head affects small-object detection accuracy relative to the other heads, backed by empirical data.
4. The proposed loss function emphasizes difficult samples in the training process, giving more weight to harder examples when calculating the regression loss. The paper claims this is an improvement over the widely used Complete Intersection over Union (CIOU) loss function. The general concept of adjusting loss functions to focus on harder examples is not new . Focal Loss was specifically designed for dealing with class imbalance and giving more weight to hard-to-detect examples. Similarly, GFL (Generalized Focal Loss) builds on this approach, placing emphasis on hard positives in object detection. The weights assigned in the Tri-Focal loss function are not well-justified in the paper. How the categorization of examples into three weights is better compared to the more fluid and general approach of Focal Loss is not clear. The hyperparameter tuning for assigning weights to different examples may also be sensitive to specific datasets, limiting the generalizability of this loss function. The authors should delve deeper into the mathematical justification for the specific three intervals used in the Tri-focal loss and provide further empirical benchmarking against loss functions like Focal Loss and EIOU under various datasets. It’s also necessary to test the Tri-focal loss on multiple datasets (not just VisDrone) to ensure it generalizes beyond specific small-object-heavy datasets.
Please clarify each issue and state how each module or innovation brings substantial advantage over prior work, supported by more theoretical or empirical comparisons.
Minor issues:
1. The introduction does a fair job contextualizing the need for advancements in small object detection for drones. However, more explicit comparisons of existing models’ performance on small objects would strengthen the motivation for this study earlier in the text. The introduction lacks a smooth transition into the specific solutions proposed and could benefit from clearer articulation of the research gap.
2. The paper reviews conventional detection algorithms (like Viola-Jones and HOG) and contrasts these with CNN-based methods like Fast R-CNN, YOLOv1, and FPN. For aerial image tasks, research like TPH-YOLOv5, DroneNet, and EL-YOLO are highlighted as attempts to deal with small object detection issues, computational complexity, and poor resolution. Related work is comprehensive but could be pruned. The citations about older methods like Viola-Jones and HOG might be unnecessary when the focus is deep-learning-based small-object detection, reducing the clutter. Also, while existing aerial object detection models are mentioned, a direct side-by-side comparison of their efficiencies (using tables or other structured presentations) would make the work more comparable. Additionally, clearer referencing styles and the inclusion of specific numeric values for accuracies/performance in related works would make the review stronger.
3. The paper concludes by reiterating the improvements ESOD-YOLO brings over YOLOv8n and emphasizes the impact that could be realized if the complexity of the model is reduced further for real-time drone edge device use. The conclusion is repetitive and lacks reflective depth in terms of limitations. No meaningful discussion is given to trade-offs in parameter numbers versus detection accuracy on smaller edge devices, especially real-time constraints.
Reviewer 5 Report
Comments and Suggestions for Authors
In this article was proposed an efficient small-object YOLO detection model (ESOD-YOLO) based on YOLOv8n for Unmanned Aerial Vehicles (UAVs) object detection. It was proposed that the Reparameterized Multi-scale Inverted Blocks (RepNIBMS) module is implemented to replace the C2f module of the Yolov8n backbone extraction network, with the objective of enhancing the information extraction capability of small objects. A cross-level multi-scale feature fusion structure, Wave Feature Pyramid Network (WFPN), is designed to enhance the model's capacity to integrate spatial and semantic information. A small-object detection head is incorporated to augment the model's ability to identify small objects. Finally, a Tri-focal loss function is proposed to address the issue of imbalanced samples in aerial images in a straightforward and effective manner.
The article is interesting and raises important issues in the discussed field.
However, it requires a few changes:
1. I think that the abstract should be shortened and limited to the most important issues raised by the article.
2. Is Figure 1 necessary, it does not contribute anything new to understanding the topic?.
3. Please add more numerical values ​​in the conclusions and describe the results obtained in more detail.
The research methodology seems correct.
The conclusions include a description of the research results. After making minor changes, the article is ready for publication.
Round 2
Reviewer 2 Report
Comments and Suggestions for Authors
The authors have revised all the issues, this paper can be accepted as it is now.
Author Response
We are extremely delighted to receive your positive feedback and the acceptance of our paper. We sincerely appreciate the time and effort you have dedicated to reviewing our work and providing valuable comments and suggestions.
Your insights and guidance have been instrumental in helping us improve the quality of our manuscript. We have strived to address all the issues raised during the review process, and we are very glad to know that our efforts have met your expectations.
We look forward to seeing our research published and contributing to the field. Once again, thank you very much for your support and for making this possible.

Reviewer 3 Report
Comments and Suggestions for Authors
1.If conditions permit, it is recommended to supplement other advanced methods for comparison
Comments on the Quality of English LanguageA small amount of editing is needed for English to make reading more natural
